

# Pyrrolizidine alkaloid variation in *Senecio vulgaris* populations from native and invasive ranges

Dandan Cheng[1], Viet-Thang Nguyen[2,3], Noel Ndihokubwayo[2,4], Jiwen Ge[5] and Patrick P.J. Mulder[6]

[1] State Key Laboratory of Biogeology and Environmental Geology, China University of Geosciences (Wuhan), Wuhan, China
[2] School of Environmental Studies, China University of Geosciences (Wuhan), Wuhan, China
[3] Faculty of Biology and Agriculture Engineering, Thai Nguyen University of Education, Thai Nguyen, Vietnam
[4] Département des Sciences Naturelles, Ecole Normale Supérieure, Bujumbura, Burundi
[5] Hubei Key Laboratory of Wetland Evolution & Ecological Restoration, China University of Geosciences (Wuhan), Wuhan, China
[6] RIKILT, Wageningen University & Research, Wageningen, The Netherlands

Corresponding author
Dandan Cheng,
dandan.cheng@cug.edu.cn,
dan-d-cheng@163.com

## ABSTRACT

Biological invasion is regarded as one of the greatest environmental problems facilitated by globalization. Some hypotheses about the invasive mechanisms of alien invasive plants consider the plant–herbivore interaction and the role of plant defense in this interaction. For example, the "Shift Defense Hypothesis" (SDH) argues that introduced plants evolve higher levels of qualitative defense chemicals and decreased levels of quantitative defense, as they are released of the selective pressures from specialist herbivores but still face attack from generalists. Common groundsel (*Senecio vulgaris*), originating from Europe, is a cosmopolitan invasive plant in temperate regions. As in other *Senecio* species, *S. vulgaris* contains pyrrolizidine alkaloids (PAs) as characteristic qualitative defense compounds. In this study, *S. vulgaris* plants originating from native and invasive ranges (Europe and China, respectively) were grown under identical conditions and harvested upon flowering. PA composition and concentration in shoot and root samples were determined using Liquid Chromatography-Tandem Mass Spectrometry (LC-MS/MS). We investigated the differences between native and invasive *S. vulgaris* populations with regard to quantitative and qualitative variation of PAs. We identified 20 PAs, among which senecionine, senecionine N-oxide, integerrimine N-oxide and seneciphylline N-oxide were dominant in the roots. In the shoots, in addition to the four PAs dominant in roots, retrorsine N-oxide, spartioidine N-oxide and two non-identified PAs were also prevalent. The roots possessed a lower PA diversity but a higher total PA concentration than the shoots. Most individual PAs as well as the total PA concentration were strongly positively correlated between the roots and shoots. Both native and invasive *S. vulgaris* populations shared the pattern described above. However, there was a slight trend indicating lower PA diversity and lower total PA concentration in invasive *S. vulgaris* populations than native populations, which is not consistent with the prediction of SDH.

## INTRODUCTION

An alien invasive plant species is a species that expands its natural range with facilitation from intentional or non-intentional human activities, tending to hazard biodiversity, ecosystem services and human well-being in its new range (*Vilà & Hulme, 2017*). Many hypotheses have been proposed to explain the invasive mechanisms of alien invasive plants (*Catford, Jansson & Nilsson, 2009*). Some explanations focus on plant-herbivore interactions and the role of plant defense. For instance, the "Enemy Release Hypothesis" (ERH) states that in a new range, introduced plants may leave behind their specialist herbivores and gain a rapid increase in distribution and abundance (*Keane & Crawley, 2002*). Loss of enemies leads to lower defense levels as plants allocate fewer resources to defense and more to growth, as according to the "Evolution of Increased Competitive Ability" (EICA) hypothesis (*Blossey & Notzold, 1995*). The "Shift Defense Hypothesis" (SDH) argues that invasive plants decrease the level of quantitative defense but increase their qualitative defense, as invasive plants still face pressure from generalist herbivores even though they escaped attack from specialists (*Doorduin & Vrieling, 2011*; *Joshi & Vrieling, 2005*; *Müller-Scharer, Schaffner & Steinger, 2004*).

Chemical defense in plants can be divided into qualitative defense and quantitative defense in relation to their effect on herbivores. Plant secondary metabolites (SMs) involved in qualitative defense are toxic to many herbivores and not very costly to produce. Those involved in quantitative defense are based on digestibility-reducing chemicals and more expensive to produce and to maintain due to the typically higher complexity of the molecules (*Feeny, 1976*; *Rhoades & Cates, 1976*). Specialist and generalist herbivores react in different ways to toxic SMs: generalist herbivores are deterred by high concentrations of toxic chemicals, while specialists are often adapted to these chemicals and use them as a cue to find their host plant. Thus, plants containing high concentrations of toxic chemicals suffer more from specialist herbivores (*Cates, 1980*). Hence, specialist and generalist herbivores inflict different selective pressures on plants, and the concentration of SMs is balanced by the opposing selective forces of specialists and generalists ("Specialist-Generalist Dilemma", *Van der Meijden, 1996*).

Moreover, different plant metabolites, even from the same groups of chemicals, may have different effects on herbivores (*Kleine & Mülller, 2010*; *Macel et al., 2005*; *Van Dam et al., 1995*). It is assumed that plants with a more diverse and/or with higher concentrations of SMs can better protect themselves when the specialist herbivores adapted to the qualitative defense chemicals are absent. Therefore, for introduced plants variation in both concentration and composition of defense chemicals is important to defend themselves against the guild of herbivores in a new range.

*Senecio* and *Jacobaea*, possessing pyrrolizidine alkaloids (PAs) as their characteristic defense compounds, have been chosen in several studies as model species to assess the

quantitative and qualitative variation in SMs in native and introduced populations. PAs act as deterrents or toxins to non-adapted herbivores and pathogens. However, specialist herbivores that are adapted to PAs can utilize them from host plants for their own benefit, such as for a food cue and oviposition (*Joosten & Van Veen, 2011*; *Macel, 2011*; *Trigo, 2011*). Higher concentrations of PAs have been found in invasive rather than native populations of *Jacobaea vulgaris* (syn. *Senecio jacobaea*; *Joshi & Vrieling, 2005*; *Lin, Klinkhamer & Vrieling, 2015*), and invasive *Senecio pterophorus* was found to have a higher concentration of PAs than its conspecific relatives (*Caño et al., 2009*; *Castells, Mulder & Perez-Trujillo, 2014*). Beside PAs from *Senecio* and *Jacobaea*, more than 350 PAs have been identified in an estimated 6,000 plants in the Boraginaceae, Asteraceae, and Leguminosae families (*Stegelmeier et al., 1999*). In this study, we selected *Senecio vulgaris* (common groundsel, Senecioneae: Asteraceae) as a model organism for the comparison of quantitative and qualitative PA variation between native and invasive populations. *S. vulgaris*, a cosmopolitan weed in temperate regions, probably originated from southern Europe (*Kadereit, 1984*), and has spread to America, North Africa, Asia, Australia and New Zealand in the 18th century (*Robinson et al., 2003*). The occurrence of *S. vulgaris* was first recorded in China in the 19th century, and it is nowadays mainly distributed in northeastern and southwestern China (*Li & Xie, 2002*; *Xu et al., 2012*). *S. vulgaris* plants of some European and Canadian populations contain high amounts (>0.6 mg/g fresh weight) of PAs (*Von Borstel, Witte & Hartmann, 1989*). *Handley et al. (2008)* investigated the invasive mechanisms of this species with respect to the interaction between plants and pathogens and the outcomes did not support the EICA hypothesis. *Zhu et al. (2017)* found that although *S. vulgari* s might have been introduced into China on multiple occasions, the Chinese populations contained smaller genetic diversity compared to European populations.

In this study, *S. vulgaris* plants from seeds collected from six native (Europe) and six invasive (China) populations were grown under identical conditions in a greenhouse. PAs were extracted from the roots and shoots of harvested *S. vulgaris* plants and measured using Liquid Chromatography-Tandem Mass Spectrometry (LC-MS/MS). According to the SDH, invasive plants tend to evolve higher levels of qualitative defense chemicals. Hence, we hypothesized that plants from invasive *S. vulgaris* populations would produce higher concentrations of PAs than those from native ranges. We also compared PA profiles in the native and invasive populations.

## MATERIALS AND METHODS

### Studies species

*Senecio vulgaris* can complete its life cycle in as little as eight weeks, producing an average of 38,300 seeds per generation and can be found in gardens, lawns, roadsides, field margins, arable lands, waste places and coastal habitats. Variation in capitula morphology, seed dormancy and growth form have been observed in different *S. vulgaris* populations (*Robinson et al., 2003*). No surveys have yet been undertaken on the amount of herbivory naturally occurring in *S. vulgaris* populations. However, it is known that *S. vulgaris* can

be the host plant of generalist herbivores such as the leafminer *Liriomyza trifolii* and the Western tarnished plant bug (*Lygus hesperus*) (*Minkenberg & Lenteren, 1986*; *Barlow, Godfrey & Norris, 1999*). The cinnabar moth (*Tyria jacobaeae*), flea beetle (*Longitarsus jacobaeae*) and ragwort seed fly (*Botanophila seneciella*) are specialists that have be used as biological control for *Jacobaea vulgaris* in North America and Australia. The first two insects have been observed also to feed on *S. vulgaris*, but it is unknown whether the ragwort seed fly can feed on *S. vulgaris*. Furthermore, a rust fungi *Puccinia lagenophorae* can infect *S. vulgaris* plants and is used as biological control of *S. vulgaris* (*Frantzen & Hatcher, 1997*). In China, we observed that leafminers and seed flies caused damage to natural populations of *S. vulgaris* and we also observed heavy herbivory by aphids on *S. vulgaris* plants grown in the greenhouse for this study. The insects have not yet been identified, and it remains to be determined whether these are specialists or not.

Some *S. vulgaris* biotypes showed increased resistance to various herbicides such as simazine, atrazine, bromacil, pyrazon, buthidazole and linuron. Therefore, *S. vulgaris* is considered as a troublesome weed, especially in horticulture where frequent cultivation occurs (*Robinson et al., 2003*). The morphology of *S. vulgaris* plants resembles that of some other *Senecio* species used as Chinese traditional medicinal plants, implicating a risk to human health if they are used as medicine or otherwise consumed by mistake (*Yang et al., 2011*).

## Plant resources, growth and harvesting

We used seeds collected from six native and six invasive *S. vulgaris* populations in Europe and China (Table 1). Achenes from six to 20 individual plants per population were kept in paper bags, air-dried and stored in the laboratory. Seeds from four to seven individuals in each population were selected for germination. Substrate made from coconut soil and sand (1:1 by volume) was placed into 12-cell boxes (size of one cell: $3.7 \times 3.7 \times 5$ cm) for seed germination. One seed was sown in each cell. After sowing, the boxes were covered with a transparent top and placed in a climate room (20 °C). The sowed seeds were watered by means of a small sprayer.

For plant rearing, we prepared substrate as described above and added slow release fertilizer (N:P:K = 14:13:13, Osmocote; The Scotts Company, Marysville, OH, USA) along with a potting medium comprising 20 g of fertilizer and 3 kg of substrate. Once 2–4 true leaves had appeared, the plants were transplanted into bigger pots (size: $8 \times 8 \times 9$ cm) containing the substrate and fertilizer and left to grow in a greenhouse.

When some of the plants began to flower, their first capitula were pruned. A week later, when the majority of plants had developed 5–10 capitula, they were then harvested. The shoots and roots were separated at their root crowns using secateurs. The shoots were rinsed using tap water. The fresh weight of the roots and shoots was separately measured. The samples were kept separately in plastic bags and then placed in liquid nitrogen prior to storage in a freezer at −80 °C. Following this, the samples were freeze-dried in an ALPHA 1-2 LD laboratory freeze-dryer (Martin Christ, Lower Saxony, Germany). The dry weight of the roots and shoots was measured before they were ground into a fine powder and

**Table 1  Sites of origin of native and invasive populations of *Senecio vulgaris*.**

| Range | Population code | Location | Coordinates | |
|---|---|---|---|---|
| Native | Barcelona | Barcelona, Spain | Lat 41.67 | Long 2.73 |
| | Pulawy | Puławy, Poland | Lat 51.40 | Long 21.96 |
| | St. Andrews | St. Andrews, UK | Lat 56.33 | Long −2.78 |
| | Fribourg | Fribourg, Switzerland | Lat 46.79 | Long 7.15 |
| | Obidos | Óbidos, Portugal | Lat 39.36 | Long −9.16 |
| | Potsdam | Potsdam, Germany | Lat 52.40 | Long 13.07 |
| Invasive | Slj.djh | Dajiuhu, Shennongjia, China | Lat 31.49 | Long 109.99 |
| | Dl.hsj | Heishijiao, Dalian, China | Lat 38.87 | Long 121.56 |
| | Lj.lsh | Lashihai, Lijiang, China | Lat 26.9 | Long 100.14 |
| | Slj.myz | Muyuzhen, Shennongjia, China | Lat 31.46 | Long 110.40 |
| | Lj.xyl | Xianyulu, Lijiang, China | Lat 26.87 | Long 100.24 |
| | Dali.sts | Santasi, Dali, China | Lat 26.70 | Long 100.15 |

homogenized using a vortex machine. Approximately 10 mg of the powder was placed into 2 mL Eppendorf tubes and stored at −20 °C until PA extraction.

## PA extraction and analysis

The extraction and analysis of PAs was performed as described in detail in our previous work (*Joosten et al., 2010*; *Cheng et al., 2011*). In brief, approximately 10 mg of the fine powdered plant material was used to extract PAs with 1 mL 2% formic acid solution in water. At a concentration of 1 $\mu$g mL$^{-1}$, heliotrine was added as internal standard to the extraction solvent. The plant extract solution was shaken for 0.5 h. Solid plant material was removed by centrifugation at 2,600 rpm for 10 min and filtered through a 0.2 $\mu$m nylon membrane (Acrodisc 13 mm syringe filter; Pall Corporation, New York, NY, USA). An aliquot of the filtered solution (25 $\mu$L) was diluted with water (975 $\mu$L) and 5 $\mu$L was injected into the LC-MS/MS system (Acquity UPLC coupled to a Quattro Premier XE tandem mass spectrometer (Waters, Milford, MA, USA)), using an Acquity BEH C18, 150 × 2.1 mm, 1.7 $\mu$m (Waters, Milford, MA, USA) UHLPC column, maintained at 50 °C, for separation of the PAs. As mobile phase A 6.5 mM ammonia in water was used and as mobile phase B acetonitrile. An analytical run was applied, starting at 100% A which was linearly changed to 50% B in 12 min, where after the mobile phase was returned to 100% A in 0.2 min. Total run time was set at 15 min and the flow was kept at 0.4 mL min$^{-1}$. Quantification of the extracts was performed against a calibration range of PA standards (0–500 ng mL$^{-1}$) in a blank plant extract. Ten analytical standards were available for quantification (Table 2). The concentrations of the remaining PAs were determined semi-quantitatively by comparison of their peak area with that of a related standard as indicated in Table 2. The limit of detection (LOD) for individual PAs in leaf tissue was approximately 0.5 $\mu$g g$^{-1}$ dry weight. LC-MS/MS analytical settings used for detection and quantification of PAs are listed in Table 2.

**Table 2  LC-MS/MS analytical settings used for detection and quantification of pyrrolizidine alkaloids (PAs).**

| No. | Pyrrolizidine alkaloid | Code | Retention time (min) | Precursor mass (m/z) | Fragment mass | Collision energy | Standard available[a] | PA used for (semi) quantification |
|---|---|---|---|---|---|---|---|---|
| 1 | Senecionine | Sn | 9.54 | 336.2 | 94.0; 120.0 | 40; 30 | Y | Senecionine |
| 2 | Senecionine N-oxide | Sn.ox | 6.68 | 352.2 | 94.0; 120.0 | 40; 30 | Y | Senecionine N-oxide |
| 3 | Integerrimine | Ir | 9.35 | 336.2 | 94.0; 120.0 | 40; 30 | Y | Integerrimine |
| 4 | Integerrimine N-oxide | Ir.ox | 6.55 | 352.2 | 94.0; 120.0 | 40; 30 | Y | Integerrimine N-oxide |
| 5 | Senecivernine | Sv | 9.79 | 336.2 | 94.0; 120.0 | 40; 30 | N | Integerrimine |
| 6 | Senecivernine N-oxide | Sv.ox | 6.75 | 352.2 | 94.0; 120.0 | 40; 30 | N | Integerrimine N-oxide |
| 7 | Retrorsine | Rt | 8.19 | 352.2 | 94.0; 120.0 | 40; 30 | Y | Retrorsine |
| 8 | Retrorsine N-oxide | Rt.ox | 5.74 | 368.2 | 94.0; 120.0 | 40; 30 | Y | Retrorsine N-oxide |
| 9 | Usaramine | Us | 7.98 | 352.2 | 94.0; 120.0 | 40; 30 | N | Retrorsine |
| 10 | Usaramine N-oxide | Us.ox | 5.62 | 368.2 | 94.0; 120.0 | 40; 30 | N | Retrorsine N-oxide |
| 11 | Seneciphylline | Sp | 8.76 | 334.2 | 94.0; 120.0 | 40; 30 | Y | Seneciphylline |
| 12 | Seneciphylline N-oxide | Sp.ox | 6.07 | 350.2 | 94.0; 138.0 | 40; 30 | Y | Seneciphylline N-oxide |
| 13 | Spartioidine | St | 8.58 | 334.2 | 120.0; 138.0 | 30; 30 | N | Seneciphylline |
| 14 | Spartioidine N-oxide | St.ox | 6.01 | 350.2 | 94.0; 138.0 | 40; 30 | N | Seneciphylline N-oxide |
| 15 | Riddelliine | Rd | 7.58 | 350.2 | 94.0; 138.0 | 40; 30 | Y | Riddelliine |
| 16 | Riddelliine N-oxide | Rd.ox | 5.20 | 366.2 | 94.0; 118.0 | 40; 30 | Y | Riddelliine N-oxide |
| 17 | Unknown N-oxide 1 | Unk1 | 4.78 | 366.2 | 94.0; 118.0 | 40; 30 | N | Riddelliine N-oxide |
| 18 | Unknown N-oxide 2 | Unk2 | 4.84 | 366.2 | 94.0; 118.0 | 40; 30 | N | Riddelliine N-oxide |
| 19 | Unknown N-oxide 3 | Unk3 | 4.88 | 368.2 | 94.0; 138.0 | 40; 30 | N | Retrorsine N-oxide |
| 20 | Unknown N-oxide 4 | Unk4 | 5.55 | 368.2 | 94.0; 138.0 | 40; 30 | N | Retrorsine N-oxide |
| 21 | Unknown N-oxide 5 | Unk5 | 5.78 | 368.2 | 94.0; 138.0 | 40; 30 | N | Retrorsine N-oxide |
| 22 | Unknown N-oxide 6 | Unk6 | 6.22 | 370.2 | 94.0; 138.0 | 40; 30 | N | Retrorsine N-oxide |
| 23 | Unknown N-oxide 7 | Unk7 | 6.57 | 402.2 | 94.0; 138.0 | 40; 30 | N | Retrorsine N-oxide |
| 24 | Unknown N-oxide 8 | Unk8 | 6.82 | 402.2 | 94.0; 138.0 | 40; 30 | N | Retrorsine N-oxide |

**Notes.**

[a] Y, standard available; N, standard not available.

## Data analysis

The Shannon index of PA diversity ($H'$) in each sample was calculated according to the formula: $H' = -\Sigma p_i * \ln p_i$, where $p_i$ is the relative abundance of each of the 20 individual PAs in a sample. The homogeneity of PA distribution in each sample (evenness, $J'$) was calculated as: $J' = H'/\ln(s)$, where s is the total number of occurring PAs in a sample. The calculation was conducted using the R package "vegan" (*Simpson et al., 2009*).

Variation in PA composition was evaluated using the concentrations of all of the 20 individual PAs detected in the shoots and roots (except usaramine N-oxide and riddelliine which were only rarely detected, see Table 3). Differences in PA composition among the populations and between the shoots and roots were evaluated using an Adonis test, a nonparametric MANOVA, in which populations and plant parts (shoots or roots) were defined as factor variables.

We visualized the variation in PA composition using a nonmetric multidimensional scaling (NMDS) method, which is analogous to a principal component analysis (PCA) or multidimensional scaling (MDS), but without distribution assumptions (*Goslee & Urban,*

Cheng et al. (2017), PeerJ, DOI 10.7717/peerj.3686

**Table 3** Pyrrolizidine alkaloids (PAs) variation in roots and shoots of *Senecio vulgaris* plants from native and invasive populations and grown under greenhouse conditions.

| No | Pyrrolizidine alkaloid | Code | PAs in roots | | | | PAs in shoots | | | | Between roots and shoots | |
|---|---|---|---|---|---|---|---|---|---|---|---|---|
| | | | Presence (%)[a] | Mean conc.[b] | Min conc. | Max conc. | Presence (%) | Mean conc. | Min conc. | Max conc. | Difference[c] ($df = 1, 58$) | Correlation[d] ($df = 1, 58$) |
| 1 | Senecionine | Sn | 100.0 | 129.1 | 2.8 | 397.6 | 100.0 | 30.9 | 1.2 | 84.7 | *** | 0.65*** |
| 2 | Senecionine N-oxide | Sn.ox | 100.0 | 1049.0 | 5.7 | 2675.2 | 100.0 | 293.9 | 2.9 | 1231.7 | *** | 0.57*** |
| 3 | Integerrimine | Ir | 100.0 | 22.6 | 0.7 | 65.9 | 100.0 | 5.0 | 0.1 | 16.6 | *** | 0.68*** |
| 4 | Integerrimine N-oxide | Ir.ox | 100.0 | 248.1 | 1.7 | 998.6 | 98.3 | 59.2 | <LOD | 242.2 | *** | 0.63*** |
| 5 | Senecivernine | Sv | 30.5 | 1.7 | <LOD | 18.0 | 18.6 | 0.4 | <LOD | 3.0 | ** | 0.68*** |
| 6 | Senecivernine N-oxide | Sv.ox | <LOD[e] | | | | | | | | | |
| 7 | Retrorsine | Rt | 94.9 | 2.5 | <LOD | 35.9 | 88.1 | 2.9 | <LOD | 63.2 | ns | 0.72*** |
| 8 | Retrorsine N-oxide | Rt.ox | 96.6 | 20.6 | <LOD | 208.8 | 94.9 | 31.6 | <LOD | 582.4 | ns | 0.45** |
| 9 | Usaramine | Us | <LOD | | | | | | | | | |
| 10 | Usaramine N-oxide | Us.ox | 1.7 | 0.1 | <LOD | 3.4 | 1.7 | 0.2 | <LOD | 12.6 | | |
| 11 | Seneciphylline | Sp | 100.0 | 11.5 | 0.4 | 63.6 | 100.0 | 17.1 | 0.3 | 83.5 | ns | 0.62*** |
| 12 | Seneciphylline N-oxide | Sp.ox | 100.0 | 92.3 | 0.9 | 376.1 | 100.0 | 161.5 | 1.3 | 1020.1 | ns | 0.50*** |
| 13 | Spartioidine | St | 93.2 | 1.8 | <LOD | 6.3 | 89.8 | 2.9 | <LOD | 17.5 | ** | 0.66*** |
| 14 | Spartioidine N-oxide | St.ox | 98.3 | 17.3 | <LOD | 57.0 | 100.0 | 29.8 | 0.4 | 212.0 | ns | 0.64*** |
| 15 | Riddelliine | Rd | 5.1 | 0.1 | <LOD | 3.4 | 1.7 | 0.1 | <LOD | 5.2 | | |
| 16 | Riddelliine N-oxide | Rd.ox | 45.8 | 0.9 | <LOD | 14.4 | 57.6 | 1.8 | <LOD | 46.1 | * | 0.48*** |
| 17 | Unknown N-oxide 1 | Unk1 | 32.2 | 0.3 | <LOD | 2.6 | 35.6 | 1.0 | <LOD | 13.5 | ns | 0.29 ns |
| 18 | Unknown N-oxide 2 | Unk2 | 61.0 | 1.0 | <LOD | 7.8 | 74.6 | 3.5 | <LOD | 32.1 | ** | 0.28 ns |
| 19 | Unknown N-oxide 3 | Unk3 | 96.6 | 9.3 | <LOD | 20.7 | 76.3 | 1.6 | <LOD | 6.7 | *** | 0.48*** |
| 20 | Unknown N-oxide 4 | Unk4 | 98.3 | 8.5 | <LOD | 27.5 | 100.0 | 30.6 | 0.7 | 148.2 | *** | 0.27 ns |
| 21 | Unknown N-oxide 5 | Unk5 | 94.9 | 18.7 | <LOD | 114.3 | 98.3 | 69.2 | <LOD | 259.1 | *** | 0.14 ns |
| 22 | Unknown N-oxide 6 | Unk6 | 88.1 | 4.5 | <LOD | 11.2 | 84.8 | 3.2 | <LOD | 19.8 | ** | 0.36* |
| 23 | Unknown N-oxide 7 | Unk7 | 44.1 | 0.6 | <LOD | 5.6 | 81.4 | 4.4 | <LOD | 33.1 | *** | 0.58*** |
| 24 | Unknown N-oxide 8 | Unk8 | 74.6 | 1.5 | <LOD | 9.1 | 86.4 | 8.3 | <LOD | 37.1 | *** | 0.53*** |
| | Total PA | | | 1641.8 | 18.4 | 4180.6 | | 758.8 | 16.3 | 2781.3 | *** | 0.58*** |

**Notes.**

[a]Presence percentage = number of root/shoot samples from which a certain individual PA was detected/number of total root/shoot sample × 100 (%).

[b]Unit of concentration: $\mu$g/g dry weight. For the PA N-oxides with unknown identity (entries 17–24) the concentrations are estimates, based on comparison of the peak area with that of riddelliine N-oxide (entries 17 and 18) or retrorsine N-oxide (entries 19–24).

[c]Difference of concentration of total PA and the individual PAs between roots and shoots was investigated by paired Wilcoxon rank tests and P-values of the tests are shown.

[d]Correlation between roots and shoots in relation to concentration of total PA and the individual PA was investigated by Spearman rank correlation tests; R and P-values of the tests are shown.

[e]<LOD: all samples below the limit of detection (0.1 $\mu$g/g dry weight).

Level of significance:

*$p < 0.05$

**$p < 0.01$

***$p < 0.001$.

Cheng et al. (2017), PeerJ, DOI 10.7717/peerj.3686

*2007*). Heatmaps were constructed to show difference between populations by using the R package "pheatmap" (*Kolde, 2015*).

We calculated the Sn/Sp ratio from the concentration of four PAs using the formula: (senecionine + senecionine N-oxide)/(seneciphylline + seneciphylline N-oxide). The ratios were square root transformed and used in a Kruskal-Wallis test to assess whether the ratios differed between populations. Between-population homoscedasticity was checked using Breusch-Pagan tests.

Total PA concentration and the individual concentrations of 20 PAs was log10 transformed and then used in analysis of PA concentration. Paired Wilcoxon rank tests were used to confirm whether the concentration of total PA and the individual PAs differed between the roots and shoots, while Spearman's rank correlation tests were conducted to investigate the correlation between roots and shoots. Breusch-Pagan tests were used to assess equality of variance between the groups. *P*-values of the results were adjusted using sequential Bonferroni method when multiple tests were performed.

To confirm whether for roots and shoots the concentration, relative abundance of individual PAs, and total PA concentration differed among populations and between ranges, nested ANOVA tests were conducted in SPSS (IBM SPSS Statistics for Windows, Version 22.0; IBM Corp., Armonk, NY, USA). Equality of variance between the groups was assessed using Levene's tests. To conduct nested ANOVA tests, we selected the 13 PAs that had an average relative abundance of more than 1%. Concentration of PAs was log transformed. Relative abundance of PAs was calculated as individual PA percentage of the total PA concentration and root square transformed.

Except nested ANOVA tests, all analyses were performed with R version 3.1.2 (*R Core Team, 2015*).

# RESULTS

## PA diversity

Of the 21 PAs reported from *S. vulgaris* in the literature, 16 PAs were included in the mass spectrometric method and detected in our samples (Fig. 1). An additional eight putative PA N-oxides, with unknown identity were detected, of which it could be ascertained, based on their protonated molecular mass, fragmentation spectra and retention times, that they were different from the 21 PAs reported previously (Table S1). In many cases, both forms of PAs (tertiary amine and N-oxide) were detected in at least part of the samples. Exceptions were senecivernine and usaramine N-oxide that were detected in a number of samples, but their counterparts senecivernine N-oxide and usaramine were below the limit of detection in all samples. Similarly, no tertiary amine counterparts of the eight unknown PA N-oxides could be identified. Thus, in total 22 PAs were detected (Table 3).

Senecionine, integerrimine, seneciphylline, and their respective N-oxides were present in the roots and shoots of all plants and all populations. Spartioidine, retrorsine and their respective N-oxides were found in all populations and in more than 90% of the individual root and shoot samples. Riddelliine N-oxide was detected in ten populations (83%), while senecivernine was detected in five populations (42%). Two PAs, riddelliine and

|  | $R_1$ | $R_2$ | $R_3$ | $R_4$ |  |  | $R_1$ | $R_2$ | $R_3$ |
|---|---|---|---|---|---|---|---|---|---|
| Senecionine | $CH_3$ | H | H | H |  | Seneciphylline | $CH_3$ | H | H |
| Integerrimine | H | $CH_3$ | H | H |  | Spartioidine | H | $CH_3$ | H |
| Senecivernine | H | H | $CH_3$ | H |  | Riddelliine | H | H | OH |
| Retrorsine | $CH_3$ | H | H | OH |  |  |  |  |  |
| Usaramine | H | CH | H | OH |  |  |  |  |  |

**Figure 1** Chemical structures of pyrrolizidine alkaloids and their corresponding N-oxides identified in *Senecio vulgaris* plants.

usaramine N-oxide, were rarely found; usaramine N-oxide was only detected in the root and shoot extracts of one plant from a native population located in Potsdam, Germany, while riddelliine was found in two plants originating from Potsdam and in one plant from an invasive population in Shennongjia, China. Six of the eight unidentified PA N-oxides were found in all populations, and three of them (Unk 3-5) were found in more than 90% of the shoot and root samples (Table 3, Tables S2–S3).

## Variation in PA composition

Overall PA diversity ($H'$) as well as evenness ($J'$) was higher in shoots than in roots, and lower in the invasive populations than in native ones (Fig. 2). Differences in PA composition were significant between organs (shoots and roots) and among populations (two factor Adonis test; organ: $df = 1$, $r^2 = 0.41$, $p = 0.005$; populations: $df = 11$, $r^2 = 0.19$, $p = 0.005$; Fig. 3). Senecionine N-oxide was the dominant component of the PA profile in the roots, followed by integerrimine N-oxide, senecionine and seneciphylline N-oxide (Fig. 4A). In the shoots, the four above mentioned PAs were also prevalent, in combination with retrorsine N-oxide and two unidentified PA N-oxides (Unk 4 and 5, Fig. 4B). The ratio between the concentration of senecionine and that of seneciphylline (Sn/Sp ratio, including the free base and N-oxide forms of these PAs) for individual plants ranged from 0.56 to 6.87; the ratio at population level was greater than 1 and differed significantly between populations (ANOVA test: $df = 11$ and 43, $F = 9.7$, $P < 0.001$, Fig. 5).

Generally, the relative abundance of individual PAs was significantly different among populations (Table 4). However, the clustering of the populations did not show any geographically related pattern (Fig. 6).

## Variation in PA concentration

Within the plants, significantly higher concentrations of senecionine, integerrimine (and their N-oxides), senecivernine and two unidentified PA N-oxides (Unk 3 and 6) were

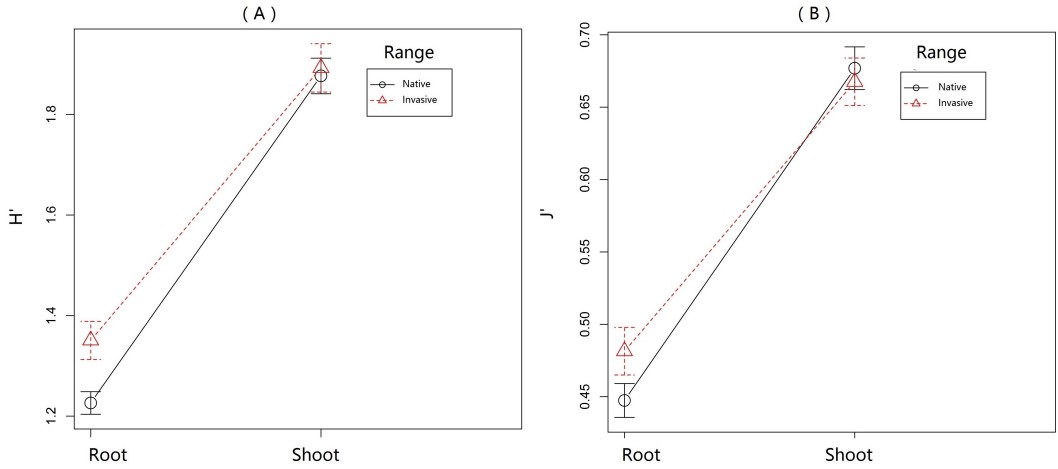

**Figure 2 Variation of pyrrolizidine alkaloids (PAs) in roots and shoots of *Senecio vulgaris* from native and invasive populations.** PA diversity was calculated as Shannon index $[H' = -\Sigma pi * \ln pi]$, where $p$ was the relative abundance of each of the 22 individual PAs in a sample. Homogeneity of PA distribution in each sample was calculated as evenness $[J' = H'/\ln(s)]$, where $s$ was the total number of occurring PAs in a sample.

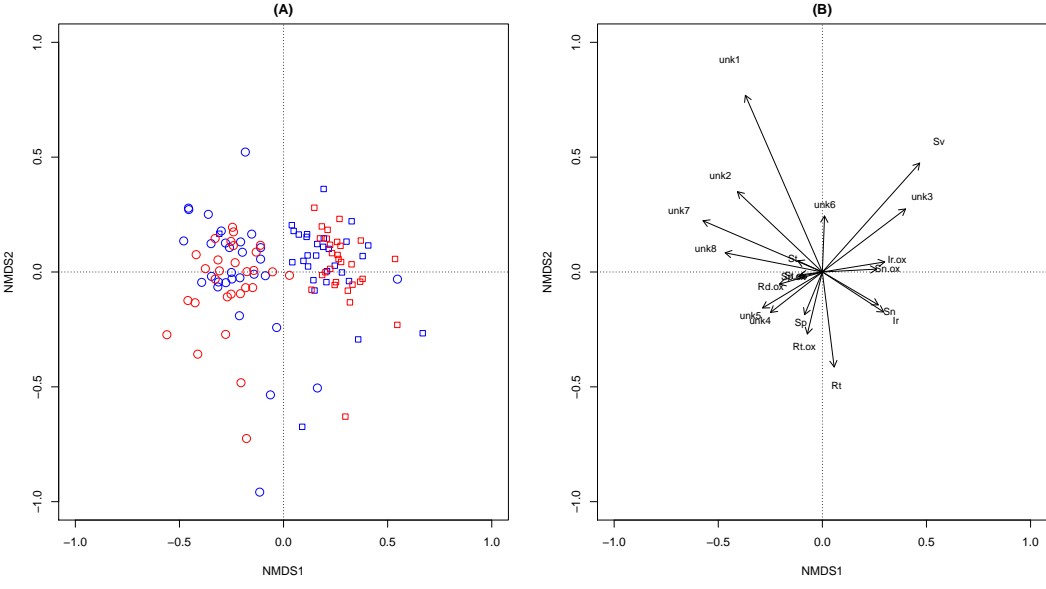

**Figure 3 Variation of pyrrolizidine alkaloids (PAs) in roots and shoots of *Senecio vulgaris* from native and invasive populations.** (A) Scoring plotting by two-dimension nonparametric multidimensional scaling (NMDS) based on concentration of 20 individual PAs. Square, roots; Dots, shoots. Red symbols were plants from invasive and the blue symbols were from native populations.(B) Loading plots of the NMDS. See details of the abbreviation of PAs in Tables 2–3.

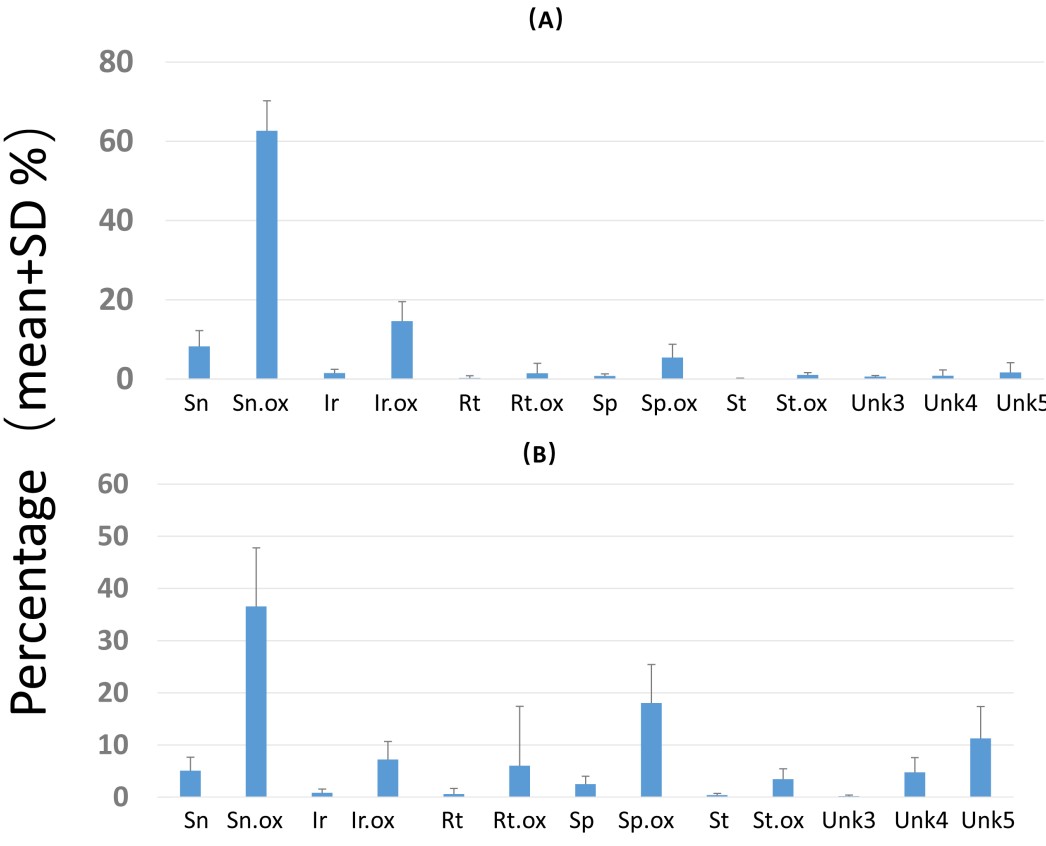

**Figure 4** **Composition of pyrrolizidine alkaloids (PAs) in roots and shoots of *Senecio vulgaris* plants.** Percentage = concentration of an individual PA/total PA concentration × 100. See details of the PAs in Tables 2–3.

present in the roots than in the shoots, but the concentrations of spartioidine, riddelliine N-oxide and five unidentified PA N-oxides (Unk 2, 4, 5, 7 and 8) were significantly lower (Table 3). The concentrations of seneciphylline, seneciphylline N-oxide, spartioidine N-oxide, retrorsine N-oxide and an unidentified PA N-oxide (Unk 1) tended to be higher in the shoots, but statistically the differences were not significant. A significant correlation between roots and shoots was found regarding the total PA concentration, as well as between most of the individual PAs, except for some unidentified PA N-oxides (Unk 1, 2, 4. 5, Table 3).

The concentration of the individual PAs and that of total PA was generally higher in plants from the native populations than in those from the invasive populations (Tables S2–S3). The difference between populations was often significant. However, significant differences between the ranges were only found for retrorsine and retrorsine N-oxide (Table 4). These two PAs were minor compounds in the PA profile of plants from both ranges (Table 3).

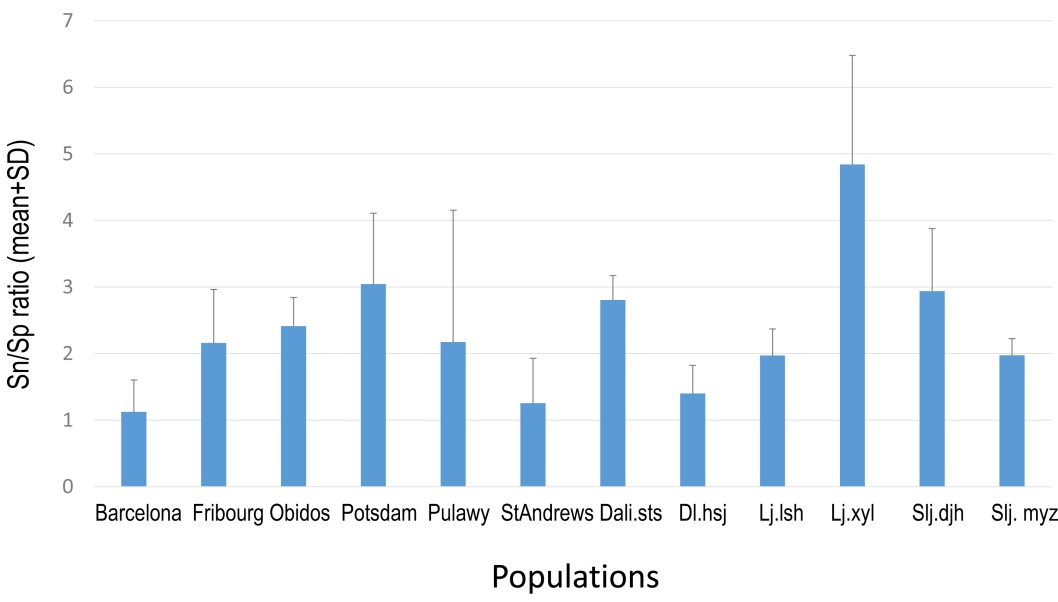

**Figure 5** **Sn/Sp ratio in shoots of *Senecio vulgaris*. plants from native and invasive populations** Sn/Sp ratio = (Senecionine + Senicionine N-oxide)/(Seneciphylline + Seneciphylline N-oxide). See details of the populations in Table 1.

## DISCUSSION

The great asset of LC-MS/MS is that it can analyse PA tertiary amine and N-oxide forms simultaneously in a single run with high sensitivity and specificity in combination with minimal sample clean-up. However, like other mass spectrometric techniques that have evolved in recent years, such as LC-QToF-MS (*Skoneczny et al., 2015*) and LC-Orbitrap-MS (*These et al., 2013*), it requires a comprehensive set of authentic analytical standards for a full quantitative result. Furthermore, although most LC-MS techniques are capable of annotating tentative PAs–based on their fragmentation spectra and or elementary composition—to establish the chemical structure of the unknowns, additional techniques, such as NMR are required.

It has been reported that PA profiles of the aboveground parts of *S. vulgaris* plants comprise seneciphylline, senecionine, retrorsine and the corresponding E-geometrical isomers, spartioidine, integerrimine and usaramine (*Hartmann & Zimmer, 1986*; *Pieters & Vlietinck, 1988*). In *S. vulgaris* PAs are primarily produced as N-oxides in the roots, which is also the dominant form of PAs in the other parts of the plant (*Hartmann & Dierich, 1998*). Apart from the PAs mentioned above, riddelliine, senecivernine, platyphylline and neoplatyphylline have been reported in the aerial parts of *S. vulgaris* plants (*Von Borstel, Witte & Hartmann, 1989*; *Yang et al., 2011*), as well as neosenkirkine (*Von Borstel, Witte & Hartmann, 1989*) and othonnine (*Xiong et al., 2012*). The 21 PAs with identified structures detected from *S. vulgaris* plants in previous studies have been summarized in Table S1 and structures of most of them were shown in Fig. 1.

Of these 21 PAs reported previously, 16 PAs were included in the mass spectrometric method, most of which were detected in this study. However, due to a lack of suitable
**Table 4   Results of the nested ANOVA tests of difference among *Senecio vulgaris* populations and ranges (native or invasive) for 13 selected pyrrolizidine alkaloids (PAs).**

| PA code | Root | | Shoot | |
| --- | --- | --- | --- | --- |
| | Population (range) | Range | Population (range) | Range |
| *Concentration of PAs[a,b]* | | | | |
| Sn | $1.24^{ns}$ | $0.17^{ns}$ | $1.12^{ns}$ | $0.37^{ns}$ |
| Sn.ox | $2.23^{*}$ | $<0.00^{ns}$ | $1.82^{ns}$ | $0.33^{ns}$ |
| Ir | $0.12^{ns}$ | $1.48^{ns}$ | $1.34^{ns}$ | $0.29^{ns}$ |
| Ir.ox | $2.23^{*}$ | $0.13^{ns}$ | $2.59^{*}$ | $0.69^{ns}$ |
| Rt | $2.98^{**}$ | $13.05^{**}$ | $3.26^{**}$ | $12.98^{***}$ |
| Rt.ox | $3.22^{**}$ | $14.15^{***}$ | $2.30^{**}$ | $14.2^{***}$ |
| Sp | $1.18^{ns}$ | $0.59^{ns}$ | $0.81^{ns}$ | $1.49^{ns}$ |
| Sp.ox | $2.23^{ns}$ | $1.26^{*}$ | $1.62^{ns}$ | $1.37^{ns}$ |
| St | $2.21^{*}$ | $2.49^{ns}$ | $2.30^{*}$ | $3.03^{ns}$ |
| St.ox | $3.14^{**}$ | $2.87^{ns}$ | $3.13^{**}$ | $3.32^{ns}$ |
| Unk3 | $1.82^{ns}$ | $0.42^{ns}$ | $2.01^{ns}$ | $2.40^{ns}$ |
| Unk4 | $3.12^{**}$ | $2.39^{ns}$ | $4.02^{**}$ | $0.49^{ns}$ |
| Unk5 | $3.02^{**}$ | $1.20^{ns}$ | $2.00^{ns}$ | $0.10^{ns}$ |
| Total PA concentration | $2.05^{*}$ | $0.11^{ns}$ | $1.81^{ns}$ | $1.48^{ns}$ |
| *Relative abundance of PAs[a,c]* | | | | |
| Sn | $0.35^{ns}$ | $1.97^{ns}$ | $2.12^{*}$ | $0.004^{ns}$ |
| Sn.ox | $2.68^{*}$ | $6.67^{*}$ | $2.10^{*}$ | $1.64^{ns}$ |
| Ir | $2.56^{*}$ | $0.16^{ns}$ | $1.46^{ns}$ | $0.51^{ns}$ |
| Ir.ox | $6.25^{**}$ | $0.09^{ns}$ | $2.95^{**}$ | $0.004^{ns}$ |
| Rt | $2.21^{**}$ | $7.96^{**}$ | $3.66^{**}$ | $2.39^{ns}$ |
| Rt.ox | $3.33^{ns}$ | $9.99^{**}$ | $5.44^{***}$ | $3.27^{ns}$ |
| Sp | $1.01^{*}$ | $0.76^{ns}$ | $1.60^{ns}$ | $0.02^{ns}$ |
| Sp.ox | $6.67^{ns}$ | $2.83^{**}$ | $2.27^{*}$ | $1.06^{ns}$ |
| St | $1.74^{***}$ | $1.25^{ns}$ | $2.83^{**}$ | $0.57^{ns}$ |
| St.ox | $6.34^{ns}$ | $15.51^{**}$ | $10.93^{***}$ | $5.09^{ns}$ |
| Unk3 | $0.73^{***}$ | $0.49^{ns}$ | $0.56^{ns}$ | $0.68^{ns}$ |
| Unk4 | $4.13^{***}$ | $1.78^{ns}$ | $2.45^{*}$ | $2.67^{ns}$ |
| Unk5 | $3.22^{**}$ | $1.39^{ns}$ | $2.05^{*}$ | $5.48^{ns}$ |

**Notes.**

[a] Nested ANVOA tests were conducted separately for each individual PA (or total PA concentration) from root and shoot samples. Concentration or relative abundance of PAs were used as independent variable, population nested in ranges ($df = 10$) and range ($df = 1$) as fixed factors. In total 59 individual plants were used, and they were from six native and six invasive populations. The relative abundance of the 13 selected PAs was at least 1%, averaged among all samples.

[b] Concentration of PAs was calculated as $\mu$g/g dry weight and log transformed for the tests.

[c] Relative abundance of PAs was calculated as individual PA percentage of total PA concentration and root square transformed for the tests.

Level of significance:

$^{ns}P > 0.05$

$^{*}p < 0.05$

$^{**}p < 0.01$

$^{***}p < 0.001$.

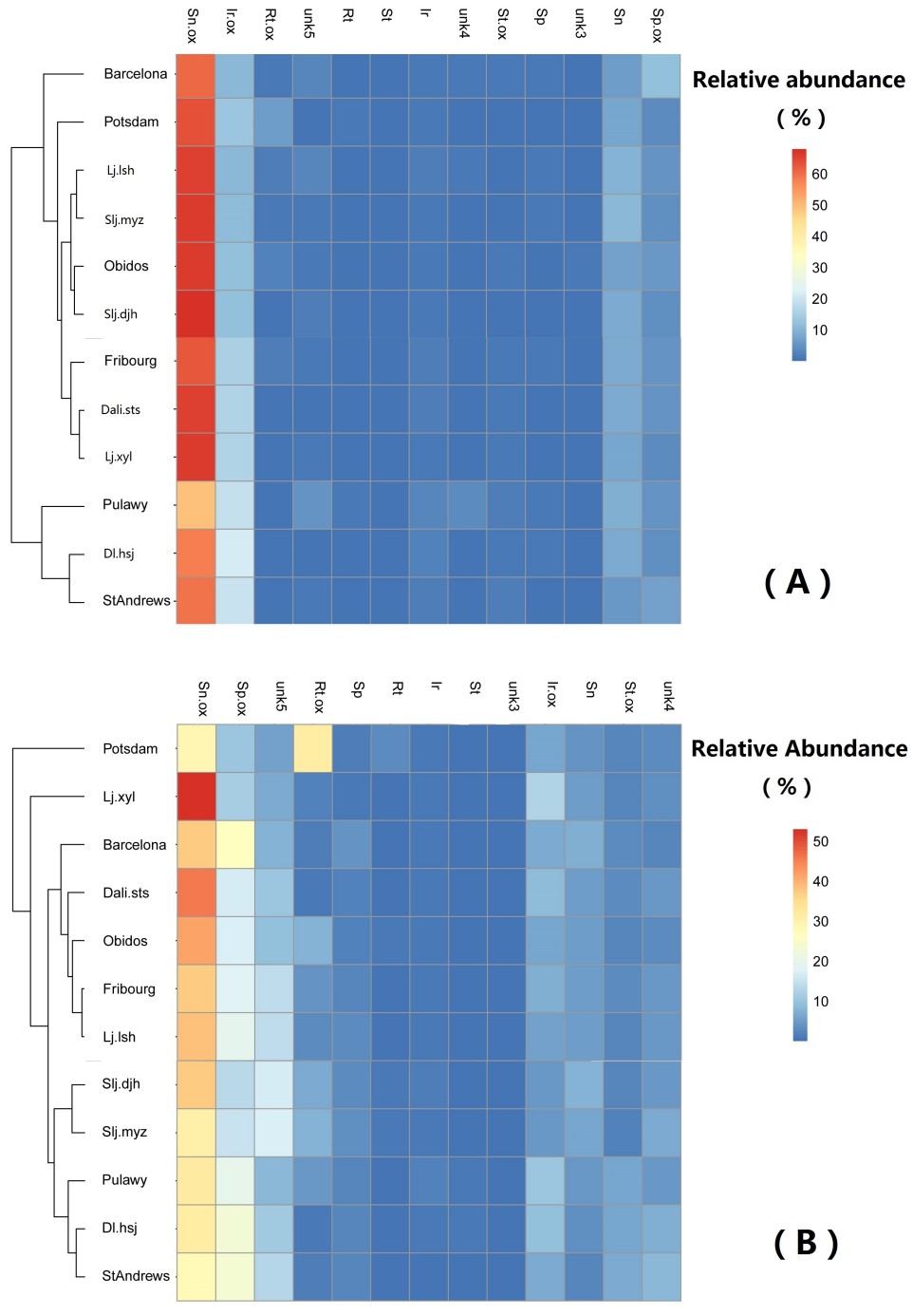

**Figure 6 Comparison of abundance of selected pyrrolizidine alkaloids (PAs) in roots and shoots of** **Senecio vulgare** **plants grown under uniform conditions in the greenhouse.** Plants were grew from seeds collected from six native and six invasive populations. Clustering algorithm and Euclidean distance metric were used on relative abundance values. See details of key to populations (at leaf of heatmap) and to PAs (on the top of heatmap) in Tables 1–3. The relative abundance of the 13 selected PAs was at least 1%, averaged among all samples.

reference standards, we were unable to search for platyphylline, neoplatyphylline, neosenkirkine, or othonnine in the root and shoot extracts of *S. vulgaris.* However, three unidentified PA N-oxides (Unk 3-5), with the same molecular mass as retrorsine N-oxide and that could be structural isomers of the latter were present in more than 90% of the samples. In particular, one PA N-oxide (Unk 5) comprised about 10% of the total PA concentration in shoot samples, although a reliable quantification of this compound due to lack of a standard could not be made. It would be worthwhile to elucidate the structure of these three PAs and explore whether they are dominant in the PA profiles of certain *S. vulgaris* plants.

Some studies have found either senecionine (*Hartmann & Zimmer, 1986*) or seneciphylline to be dominant (*Lüthy, Heim & Schlatter, 1983*), while others have found both PAs to be dominant in *S. vulgaris* (*Von Borstel, Witte & Hartmann, 1989*; *Brown & Molyneux, 1996*). In this study, senecionine was generally present in higher concentrations than seneciphylline.

The shoots and roots of *S. vulgaris* plants differed in that shoots showed more divergent PA profiles and that the shoots had a lower total PA content than the roots. Although there were significant differences in PA variation between the shoots and roots, these parts were positively correlated regarding the concentrations of total PAs and most of individual PAs (Table 3). This pattern could be explained by the processes of PA synthesis and accumulation in *S. vulgaris* plants, as PAs are primarily produced as senecionine N-oxide in the roots, while structural transformation mainly occurs in the shoots. Usually there is little turnover of PAs once being produced and they translocate to plant tissues mainly via the phloem (*Hartmann & Dierich, 1998*). Similar patterns regarding differences and correlations of PAs between the roots and shoots have been found in *J. vulgaris* (*Cheng et al., 2011*; *Joosten et al., 2011*).

Higher PA concentrations in the belowground parts compared with the aboveground parts of *S. vulgaris* plants have been found in the vegetative stage. It has been reported that when the plants have produced buds, the highest PA concentrations are found in the capitula, while the stems and leaves generally contain lower total PA concentrations compared to the roots (*Hartmann & Zimmer, 1986*). This consistent with our finding that the total PA concentration was lower in the shoots than in the roots when the *S. vulgaris* plants were not yet flowering.

The indexes of PA diversity and evenness were somewhat lower for plants from invasive ranges than those from the native range (Fig. 2). This indicated that invasive *S. vulgaris* populations tended to produce less diverse PA profiles than the native ones. However, this trend is much weaker than observed in some other invasive species. For instance, native *J. vulgaris* populations expressed four chemotypes (*Macel, Vrieling & Klinkhamer, 2004*), while in invasive *J. vulgaris* populations one chemotype dominated (*Joshi & Vrieling, 2005*). PA diversity in *S. pterophorus* (native to South Africa) was reduced after introduction in Europe and Australia (*Castells, Mulder & Perez-Trujillo, 2014*). Furthermore, invasive *Tanacetum vulgare* plants contained a smaller number of qualitative defense compounds than the native ones (*Wolf et al., 2011*).

We also found that invasive *S. vulgaris* populations did not produce higher concentrations of individual and of total PAs than native populations. These results did not agree with our prediction deduced from the SDH. Some studies have found that PA levels of related species significantly increased in the invaded range. For example, invasive populations of *S. inaequidens*, *S. pterophorus* and *J. vulgaris* all showed a significantly higher total PA concentration than their native conspecifics (*Joshi & Vrieling, 2005*; *Caño et al., 2009*; *Castells, Mulder & Perez-Trujillo, 2014*; *Lin, Klinkhamer & Vrieling, 2015*). Some other invasive species appear to have evolved towards decreased chemical defense levels but they may have developed other compensatory mechanisms that contribute to their invasion success. For instance, invasive genotypes of *Sapium sebiferum* evolved a reduced defense and resistance ability, but were more tolerant and outperformed the native genotypes under higher levels of herbivore attack (*Zou et al., 2008*).

The prerequisite of the EICA hypothesis and SDH is that invasive plants face a lower specialist herbivore pressure in invasive ranges. We could confirm that *S. vulgaris* populations in China were attacked by insect herbivores but we did not determine whether the insects were specialists or not. Although it is likely that most insects will be generalists, it is not impossible that there may be one or more specialists among them that have adapted to *S. vulgaris*, since *S. vulgaris* has a long invasive history and more than 60 congeneric species have been identified in China (*Chen, 1999*). Since there are significant variations between populations, a good revisiting study on the EICA hypothesis and SDH needs enough populations for a robust statistical analysis, and it is also important to describe and cluster invasive populations by analysis of their genetic structure in the different ranges; otherwise it remains difficult to determine whether the differences between native and invasive populations are the result of evolution or of pre-adaption (*Pan et al., 2013*; *Turner, Hufbauer & Rieseberg, 2014*; *Siemann et al., 2016*; *Schrieber et al., 2016*).

Taking into account the high PA levels present in *S. vulgaris* and the toxic effect that PAs exert on most herbivores, it reasonable to assume that PAs play an important role in the chemical defense of *S. vulgaris*. However, there are also other metabolites that can function as chemical defense in *S. vulgaris*. For instance, an oplopane sesquiterpene and jacaranone were identified from *S. vulgaris* (*Liu, Zhang & Wang, 2010*). Both compounds (or similar compounds) have a negative effect on insect feeding (*Lajide, Escoubas & Mizutani, 1996*; *Reina et al., 2001*; *Xu, Zhang & Casida, 2003*). It will be interesting to investigate whether the levels of other qualitative defense compounds such as oplopane sesquiterpenes and jacaranone are higher in invasive *S. vulgaris* populations than in native ones, as the SDH would predict. It may be advantageous to use a non-targeted analysis approach to explore for metabolites of potential significance, as was recently shown in the study of *Skoneczny et al. (2017)*.

## CONCLUSIONS

As the *Senecio vulgaris* plants from native and invasive ranges were grown under identical conditions, the differences in PA concentration and PA composition between ranges and between populations might thus be explained by their genetic variation. In our study

the invasive *S. vulgaris* populations had slightly less diverse PA profiles and tended to have lower concentrations of individual PAs compared to the native populations. This finding is in contrast to the predictions of the SDH. However, the current findings should also be treated with caution given the limited number of populations sampled, the lack of background information about herbivore guilds feeding on *S. vulgaris* and the limited knowledge on the genetic structure of *S. vulgaris* populations in the different ranges. Future studies should focus on sampling a larger number of populations and screening for a wider array of plant metabolites in order to address these questions.

## ACKNOWLEDGEMENTS

We are very thankful to K Vrieling, T de Jong, E Castells, H Müller-Schärer, J Joshi, R Abbott, G Korbecka, X Wei and X Yang for seed collecting. J Li, L Feng, Y Shi and X Deng are thanked for their help in plant rearing and harvesting.

### Funding

This work is supported by the Fundamental Research Funds for the Central Universities (CUG130411) and National Natural Science Foundation of China (31570537 and 31200425) granted to Dandan Cheng. Viet Thang Nguyen and Noel Ndihokubwayo are supported by the China Scholarship Council (CSC) for the study in China. The funders had no role in study design, data collection and analysis, decision to publish, or preparation of the manuscript.

### Grant Disclosures

The following grant information was disclosed by the authors:
Fundamental Research Funds for the Central Universities: CUG130411.
National Natural Science Foundation of China: 31570537, 31200425.
The China Scholarship Council (CSC).

### Competing Interests

The authors declare there are no competing interests.

### Author Contributions

- Dandan Cheng conceived and designed the experiments, performed the experiments, analyzed the data, contributed reagents/materials/analysis tools, wrote the paper, prepared figures and/or tables, reviewed drafts of the paper.
- Viet-Thang Nguyen performed the experiments, analyzed the data, wrote the paper, prepared figures and/or tables.
- Noel Ndihokubwayo performed the experiments.
- Jiwen Ge reviewed drafts of the paper.
- Patrick P.J. Mulder conceived and designed the experiments, performed the experiments, analyzed the data, contributed reagents/materials/analysis tools, reviewed drafts of the paper.
## Data Availability

The raw data has been supplied as Supplemental dataset.

## Supplemental Information

Supplemental information for this article can be found online at http://dx.doi.org/10.7717/peerj.3686#supplemental-information.

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
