# Peer review of "Pyrrolizidine alkaloid variation in Senecio vulgaris populations from native and invasive ranges"

_PeerJ, doi:10.7717/peerj.3686_

## Round 0.1 · original submission · Major Revisions

Dear Dr. Cheng,

Your manuscript “Pyrrolizidine alkaloid variation in Senecio vulgaris populations from native and invasive ranges”, which you kindly submitted to PeerJ for publication has been reviewed by two competent colleagues and their comments are enclosed.

As you can read, the reviewers expressed significant concerns with respect to the experimental design (number of samples) and presentation of data. Although the reviewers found your approach interesting, the manuscript needs substantial improvements (major revision).

The most important concerns are related to the description of the method, the number of PAs detected in relation to the known ones from literature and – most importantly – the evaluation and presentation of the data sets. Here, I highly recommend to follow the advice of the second reviewer and to include proper biostatistical analyses using heat map and principal component analyses.

To land your manuscript in PeerJ, you have to perform a major revision by addressing all(!) points raised by the reviewers.

Best regards
Bettina Hause

Reviewer 1 ·

Basic reporting

Referee comments on the article titled:
Pyrrolizidine alkaloid variation in Senecio vulgaris populations from native and invasive ranges. (# 17300)
Basic Reporting:
This article has been well written and provides a comparative study of quantitative and qualitative variation pyrrolizidine alkaloids (PAs) of six species of native and introduced Senecio vulgaris from Europe and China. The manuscript is well written,
The results could be improved with the study of a greater number of species of native and introduced S. vulgaris. Future studies should focus on this aspect.
-Clear and profesional English used. Introduction is correct and literature well referenced.
-The data obtained are good and constitute a good contribution to the literatura.
- All methods described are given in sufficient detail: sample preparation, extraction and detection techniques, with the same methodology in described Phytochemical Analysis (Lotte Joosten and P.P.J. Mulder et al., 2010, 21, 197-204). Curiously, this reference is not quoted in this article. It should be included in the manuscript.
-A figure of structures of isolated pyrrolizidine alkaloids: senecionine, integerrimine, senecivernine, spartioidine, riddelliine, seneciphylline, spartioidine, retrorsine and their N-oxides would be convenient to add to the manuscript.
-The conclusions are well-defined, although very limited by the results obtained, since a broader sampling would have to be done.
On line 208 the authors say:
Of the 21 PAs reported in the literature 16 PAs were included in the mass spectrometric method and detected in our samples. Based on their protonated molecular mass and fragmentation patterns, eight putative PA N-oxides, with unknown identity were detected, of which it could be ascertained that they were different from the 21 PAs reported previously (Table S1).
Only 21 PAs reported in the literature????
Because more than 660 PAs and PA N-oxides have been identified in over 6,000 plants (Stegelmeier BL, et al.., 1999. Pyrrolizidine alkaloid plants, metabolism and toxicity. J. Nat. Tox. 8:95–116.).

Experimental design

Methods should be described with sufficient information to be reproducible by another investigator. With the same methodology in described Phytochemical Analysis (Lotte Joosten and P.P.J. Mulder et al., 2010, 21, 197-204).

Validity of the findings

Results are welcome, complete the existing ones and the literature

Additional comments

The conclusions are not very significant because the sampling should have been broader.

·

Basic reporting

Basic reporting is clear and concise. Literature review is reasonable. You might wish to evaluate the pyrrolizidine alkaloid paper published by Skoneczny et al in 2015 with Echium spp. regarding methods and use of LC/MS QToF analysis. This has been reported and additional papers are now in review using similar methodology. Use of English language is reasonable. Hypotheses generated are reasonable.

Experimental design

The experimental design was generally reasonable. However, the authors could have benefited from use and application of metabolomics to explain a complex dataset with more appropriate bioinformatics approaches. I might have suggested the authors use targeted analysis for PA and non-targeted analysis for other metabolites of potential significance to invasion ecology. The authors might have designed an optimum method for chromatographic separation and ID of these metabolites using both positive and negative ion modes. I would refer to papers published by Skoneczny again in Molecules 2017 which shows that more than one family of metabolites may be critical in plant invasion. As most plant species are highly conserved for the biosynthesis of key metabolites, likely differences among populations may be limited, and if significance is noted, it may be explained by environmental or G X E differences. When one set of constituents is upregulated, another set is often down regulated as plants typically have a limited pool of resources to synthesize plant metabolites, or defense metabolites. I think it would be useful to think about the implications for metabolomics approaches in the future. I would have liked to see PCA applied for each treatment so one could see the impact of population on the PCA scattergram. It seems that there is much variation in results associated with each population treatment but data would be clearer if represented in this manner. The use of heat maps would also simplify the presentation of the data for each of the 20 PAs in a heatmap as compared by population as well. Please try to see if in fact you could present your data using such clear bioinformatics approaches which help the reader visually understand a complex data set. It is disappointing that you did not find greater inherent production of these metabolites in the invasive plant populations in a common garden experiment but what may actually be more relevant is what levels of these metabolites were actually observed in the same location in field, in situ rather than in common garden experiments or in contrast to common garden experiments. Sometimes what is happening in the field is of greater significance. So sampling from each site in the field would have also been a good idea.

Validity of the findings

Data is generally robust but I would like to see the data re presented by analysis using heat maps and principal componenet analysis or other more visual approaches to data presentation rather than large tables. These are more user friendly and help to clarify your points regarding individual metabolites to the reader. It would also have been interesting if you presented trends associated with only top 3 key PAs for example in most populations......or top 5 to see if any trends emerge. The data did not fit the hypothesis outlined but perhaps other hypotheses should be queried. If the authors had actually surveyed plants by use of metabolomics from each of these field sites, findings may also present a totally different data set than that created by common garden experiments.

Additional comments

Paper is interesting and generally well written but I would like to see more graphical presentation of large data sets rather than use of difficult to interpret tables. Please see other more recently published papers using LC/MS QToF or other LC/MS techniques and see if you might apply these to your own dataset to improve readability for a general audience.

---

## Round 0.2 · accepted · Accept

Dear Dr. Cheng,

Your manuscript “Pyrrolizidine alkaloid variation in Senecio vulgaris populations from native and invasive ranges”, which you kindly re-submitted to PeerJ for publication has been reviewed again by one of the former reviewers. Additionally, I went carefully through your revised version and checked your responses to reviewer 2.

As you can read, the reviewer is now completely satisfied with your current manuscript. I do agree to this opinion, you addressed convincingly all points raised by the reviewers concerning the former version. Therefore, it is a pleasure for me to inform you that your manuscript is accepted as it stands.

Best regards
Bettina Hause

Reviewer 1 ·

Basic reporting

This article has been well written and provides a comparative study of quantitative and qualitative variation pyrrolizidine alkaloids (PAs) of six species of native and introduced Senecio vulgaris from Europe and China.
-Clear and profesional English used. Introduction is correct and literature well referenced.
-The data obtained are good and constitute a good contribution to the literature.
- All methods described are given in sufficient detail: sample preparation, extraction and detection techniques, with the same methodology in described Phytochemical Analysis (Lotte Joosten and P.P.J. Mulder et al., 2010, 21, 197-204).
-A figure of isolated pyrrolizidine alkaloids: senecionine, integerrimine, senecivernine, spartioidine, riddelliine, seneciphylline, spartioidine, retrorsine and their N-oxides has been added to the manuscript.
-The conclusions are well-defined.

Experimental design

-Research question well defined, relevant & meaningful.

-Methods described with sufficient detail to be reproducible by another investigator.

Validity of the findings

The data obtained are good and constitute a solid scientific contribution.


- The conclusions are well defined, but limited, but opens the possibility for future work in the same line of research.

Additional comments

Excellent article, although it could be very interesting to identify unknown PAs